# Development of an In Vivo Extended-Spectrum Cephalosporin-Resistant *Escherichia coli* Model in Post-Weaned Pigs and Its Use in Assessment of Dietary Interventions

**DOI:** 10.3390/ani13060959

**Published:** 2023-03-07

**Authors:** Tanya Laird, David Jordan, John Pluske, Josie Mansfield, Stuart Wilkinson, David Cadogan, Sam Abraham, Mark O’Dea

**Affiliations:** 1Antimicrobial Resistance and Infectious Diseases Laboratory, Harry Butler Institute, Murdoch University, Murdoch, WA 6150, Australia; 2NSW Department of Primary Industries, Wollongbar, NSW 2477, Australia; 3Agricultural Sciences, Harry Butler Institute, Murdoch University, Murdoch, WA 6150, Australia; 4Faculty of Sciences, The University of Melbourne, Parkville, VIC 3052, Australia; 5Australasian Pork Research Institute Ltd., Willaston, SA 5118, Australia; 6Feedworks Pty Ltd., Romsey, VIC 3434, Australia

**Keywords:** antimicrobial resistance, postbiotics, swine

## Abstract

**Simple Summary:**

Antimicrobial resistance in pork production has led to investigations and analyses of alternate control strategies against bacterial infection. This study developed and assessed an in vivo model for analysis of several strategies in reducing the carriage of extended-spectrum cephalosporin-resistant *Escherichia coli* in weaner pigs. More specifically, the study aimed to determine the potential of postbiotics, the fermentation products of probiotic strains, as a control strategy against the carriage of extended-spectrum cephalosporin-resistant *Escherichia coli*. The model was demonstrated to successfully colonise weaner pigs with extended-spectrum cephalosporin resistant *Escherichia coli*. A strong tendency was observed for a reduction in antimicrobial-resistant *Escherichia coli* due to inclusion of some postbiotics in the diets. However, further investigation into these postbiotics is suggested using an increased sample size. Overall, the established model offers a method for the analysis of alternate control strategies and their effects on antimicrobial-resistant bacteria. This is pivotal in the development and establishment of antimicrobial-resistant bacteria control strategies within the livestock sector.

**Abstract:**

Current interventions targeting antimicrobial resistance (AMR), a major impact on commercial pork production, focus on reducing the emergence of AMR by minimising antimicrobial usage through antimicrobial stewardship and a range of alternative control methods. Although these strategies require continued advancement, strategies that directly aim to reduce or eliminate existing antimicrobial resistant bacteria, specifically bacteria resistant to critically important antimicrobials (CIAs), need to be investigated and established. This study established an in vivo model for examining the effects of postbiotics, in the form of *Lactobacillus acidophilus* fermentation products (LFP) and *Saccharomyces cerevisiae* fermentation products (SFP), on the shedding of extended-spectrum cephalosporin (ESC)-resistant *E. coli*. The model was successful in demonstrating the presence of ESC-resistant *E. coli* as evidenced by its detection in 62 of 64 pigs. There was a strong trend (*p* = 0.065) for the SFP postbiotics to reduce the shedding of ESC-resistant *E. coli*, indicating positive impacts of this additive on reducing the carriage of bacteria resistant to CIAs. Overall, this in vivo model enables future evaluation of strategies targeting ESC-resistant *E. coli* while increasing our knowledge on the carriage of ESC-resistant *E. coli* in pigs.

## 1. Introduction

Antimicrobial resistance (AMR) continues to be one of the greatest public health concerns in today’s society. It is predicted to cause 10 million deaths per year by 2050 if left uncontrolled [1]. Of particular concern is the more recent emergence and dissemination of resistance towards critically important antimicrobials (CIAs). These antimicrobials are reserved as the last line of defence in life-threatening human infections, with CIA resistance rendering antimicrobial therapy ineffective [2]. Bacteria resistant to CIAs, including extended-spectrum cephalosporins (ESCs) and fluoroquinolones, have been detected globally in food-producing animals, amplifying the risk of further spread between animals and into humans, via direct or indirect transmission [3,4,5]. The presence of this resistance is therefore of a One Health concern and urgently requires action.

Extended-spectrum cephalosporin-resistant *E. coli* was first detected in food-producing animals in 1996 and despite tightened regulations of ESC usage, has since been detected globally in swine [3,6,7,8]. This resistance is predominantly attributed to variants of the *bla*_CTX-M_ and *bla*_CMY_ genes [5]. Two factors heightening the threat of ESC-resistance is the ease of transmission through plasmid carriage, as demonstrated by the global presence of the highly transferable IncI1-*bla*_CTX-M-1_ plasmid, and its long-term persistence in the absence of direct selection pressures [6,9,10]. Although many prophylactic approaches have been implemented to reduce the emergence and dissemination of AMR, there are limited strategies directly targeting AMR, specifically ESC-resistance, and its emergence in pork production systems.

There is an array of strategies to reduce antimicrobial-resistant *E. coli* carriage including bacteriophage therapy, antimicrobial peptides and postbiotics [11,12,13]. Postbiotics, the mixture of bioactive compounds resultant from the fermentation of probiotic strains, have been reported to reduce levels of AMR [14]. These authors challenged broilers with a multi-drug resistant *Salmonella* strain, with broilers supplemented with *Saccharomyces cerevisiae* fermentation product (SFP) demonstrating a significant reduction in the percentage of *Salmonella* resistant to chloramphenicol. Postbiotics have also been reported to increase host health and protect against enteric pathogens through modulation of the gut microbiome [13,15]. Although studies have reported conflicting impacts on weight gain in livestock offered *Lactobacillus acidophilus* fermentation product (LFP) and SFP, an increased diversity of the faecal microbiome and reduced pathogen levels in ileal mucosa have been reported in swine [16,17]. The effect of postbiotics on reducing levels of CIA-resistant bacteria in pigs requires further investigation, with this potentially minimising the threats of ESC-resistant *E. coli* while providing additional host health benefits.

An important aspect of examining alternative dietary strategies to reduce resistance levels in commensal bacteria is the development of a suitable and reliable in vivo experimental model, rather than evaluations being confined to in vitro experiments [11]. A novel model assessing the in vivo clearance of ESC-resistant *E. coli* in weaner pigs was developed as part of this study. This model involved challenge with an ESC-resistant *E. coli* (commensal strain) and application of the high-throughput Robotic Antimicrobial Susceptibility Platform (RASP) [18] for quantification of ESC-resistant *E. coli* in individual pigs at multiple timepoints. We hypothesised that pigs receiving feed supplemented with LFP or SFP, alone or in combination, would demonstrate superior clearance of ESC-resistant *E. coli* in weaner pigs challenged with ESC-resistant *E. coli*.

## 2. Materials and Methods

This experiment was approved by the Murdoch University Animal Ethics Committee (R3181/19).

### 2.1. Animals, Housing and Experimental Design

Dietary treatment groups were allocated to pens by a randomised block design with four replicate pens of each treatment. Pigs were recruited to the trial from a high-health-status Australian herd. Sows and gilts were vaccinated with ECOvac (MSD Animal Health, Bendigo, Australia) and PLEvac (MSD Animal Health, Bendigo, Australia). Weaner pigs recruited for the study were removed prior to receiving any vaccinations. Only male pigs were available as per the farms breeding programme. The 64 male piglets (Large White x Landrace) were weaned at ~21 days of age and moved from a commercial piggery in Western Australia to the animal housing facility at Murdoch University.

Allocation of pigs to pens was based on entry weight with each pen housing 4 pigs resulting in 16 pigs representing each treatment group. All pens were equipped with a 5-space feeder, a nipple drinker, plastic bottles for enrichment, and were constructed of metal with plastic flooring. Pigs received feed and water ad libitum. Pigs were acclimatised for 7 days before inoculation with ESC-resistant *E. coli*, with this day designated as day 0. All pigs were weighed on arrival (day −7), with pigs and feed subsequently weighed on day 0 and then 7, 14, 21 and 28 days thereafter for determination of performance data. Pigs were therefore kept for a total period of 35 days.

Experimental diets, formulated to meet the energy and nutrient requirements of these pigs, comprised a control diet (CON), CON supplemented with 2000 ppm LFP (LFP), CON supplemented with 2000 ppm SFP (SFP), and CON supplemented with the combination of 2000 ppm of LFP and 2000 ppm SFP (LAS) (Table 1) (Appendix A). The wheat- and barley-based diets comprised a mixture of vegetable and animal protein sources typical for a diet fed to weaner pigs in Western Australia and were formulated to contain 10.1 MJ/kg of net energy, 13.5 g/kg of standardised ileal digestible lysine, and 205 g/kg crude protein. Diets were fed in meal form. Fermentation products were Diamond V SynGenX™ (Feedworks, Romsey, Australia) and Diamond V Original XPC™ (Feedworks, Romsey, Australia) for LFP and SFP, respectively.

### 2.2. ESC-Resistant E. coli Inoculation

All pigs were treated with 25 mg (50 mg/mL) ceftiofur via intramuscular injection on day −1 (i.e., 6 days after weaning) to promote colonisation with ESC-resistant *E. coli* upon inoculation. Pigs were inoculated with ESC-resistant *E. coli* strain SA13 [9] on days 0 and 1 (i.e., days 7 and 8 after weaning) using gelatin capsules for delivery as described previously [17], with the exception of the original strain grown on CHROMagar™ ESBL (MicroMedia, Edwards Group, Narellan, New South Wales, Australia) and a single, blue colony being recultured. Strain SA13 is a commensal *E. coli* strain with a IncI1-*bla*_CTX-M-1_ plasmid. On day one and two of inoculation, pigs received two capsules containing 1.92 × 10^8^ colony forming units (CFU) per capsule.

### 2.3. Faecal Sampling and Processing

Rectal swabs were collected from each pig on days −1, 1, 2, 3, 5, 7, 14, 21 and 28 of the experiment. The average weight of the swab end was calculated by averaging the weight of 10 clean swab ends. Following rectal swabbing, all sample swab ends were cut, weighed and then suspended in 15 mL centrifuge tubes containing 5 mL of PBS.

### 2.4. RASP Quantification

Samples were placed onto the RASP for quantification of ESC-resistant *E. coli* and total *E. coli* using CHROMagar™ ESBL and CHROMagar™ ECC (MicroMedia, Edwards Group) agar plates, respectively, as previously described [17,18,19]. Briefly, samples were diluted to the expected concentrations with two dilutions plated onto each agar plate using dual spiral plating (Figure 1). Agar plates were incubated overnight at 37 °C and placed back onto the RASP system for imaging of plates. Repeated plating at different dilutions was completed if colony density was outside of the dilution range, with quantification determined from repeated plates. Colonies were counted manually from captured images with the distinction between species based on colour of colonies on chromogenic agar.

### 2.5. Statistical Analysis

Statistical analysis and graphing were conducted using STATA (v15.1) and R Studio (v1.2.5033). Liveweight (kg) of pigs from day −7 to 28 was first analysed descriptively to assess the form of temporal trends and generalised additive models were used to fit smoothing splines to non-linear trends initially with pigs, pens and rooms as random effects in a full model. Simpler models were assessed for suitability based on the Akaike information criteria and the final model used to produce estimates of the mean effect of diet on liveweight through the experimental period with 95% confidence intervals (CI) relied on for interpreting the impact of sampling error on differences. Bacterial quantification data were log transformed and analysed across time for ESC-resistant *E. coli* and total *E. coli*, termed ESC and ECC shedding density, respectively. A one-way ANOVA followed by Tukey post hoc test was used to analyse bacterial shedding and performance data. Statistical significance was accepted at *p* < 0.05, and a trend was recognized at *p* < 0.1.

## 3. Results

### 3.1. ESC-Resistant E. coli Quantification

No ESC-resistant *E. coli* were detected prior to challenge. The concentration of ESC-resistant *E. coli* was highest on day 1 (i.e., 24 h after the first ESC-resistant *E. coli* inoculation) across all treatment groups (Figure 2) (Appendix A). This was lowest in the LAS group, at 3.4 log_10_ CFU/g, in comparison to 4.0, 4.1 and 4.1 log_10_ CFU/g, in the SFP, LFP and control groups, respectively, but there was no overall statistical difference between diets (*p* > 0.05). A second peak in ESC-resistant *E. coli* was detected in all groups occurring on day 7 for the LAS and SFP groups and day 14 for the LFP and control groups. The concentration of ESC-resistant *E. coli* showed a trend (*p* < 0.1) between the SFP and CON groups with the SFP group demonstrating a reduced ESC-resistant *E. coli* concentration. No difference in the concentration of ESC-resistant *E. coli* was detected at the final time point, with ESC-resistant *E. coli* concentration ranging from 0.1 log_10_ CFU/g in the SFP group to 0.2, 0.4 and 0.5 log_10_ CFU/g in the control, LAS and SFP groups, respectively.

The bacterial shedding density (a measure of bacterial shedding across the full duration of the study) was used to statistically analyse the shedding of ESC-resistant *E. coli* and total *E. coli* (Figure 3). The mean (with 95% CI) ESC shedding density was highest in the control group, 47.3 [30.8, 63.7], compared to 28.7 [12.2, 45.1], 30.6 [14.2, 47.1] and 35.9 [19.5, 52.3] for the SFP, LAS and LFP groups, respectively, but these were statistically similar (*p* > 0.05) (Figure 3).

Total *E. coli* shedding was similar (*p* > 0.05) between treatment groups with the mean shedding density ranging from 141.9 (123.0, 160.7) in the LAS group to 166.5 (147.7, 185.3) and 173.8 (155.0, 192.6) in the control and LFP groups, respectively (Figure 3).

The number of pigs with ESC-resistant *E. coli* was highest in all treatment groups on day 1 (i.e., 24 h after the first ESC-resistant *E. coli* inoculation). This declined by 48 h post-challenge with only 38% of pigs harbouring ESC-resistant *E. coli* on day 2 at detectable levels. The second peak detected in the quantification of ESC-resistant *E. coli* was also reflected in the percentage of pigs harbouring ESC-resistant *E. coli*, with an increase from 13 to 69% of pigs in the control group with ESC-resistant *E. coli* on day 5 and 14, respectively (Figure 4). Over the entire study, ESC-resistant *E. coli* was undetected in two pigs. Meanwhile, only nine samples had no *E. coli* detected when grown on ECC agar.

### 3.2. Abundance of ESC-Resistant E. coli Relative to Total E. coli

The abundance of ESC-resistant *E. coli* relative to the total *E. coli* population demonstrated similar trends to the ESC-resistant *E. coli* concentration (Figure 5) (Appendix A). This was evident in the first peak of abundance of ESC-resistant *E. coli* followed by a second peak on days 7 or 14 depending on the treatment group. The lowest abundance of ESC-resistant *E. coli* occurred on the final day of the trial demonstrating a natural clearance.

### 3.3. Pig Performance

Pig live weight increased over the duration of the study, starting at an average of 6.62 kg (day −7) across all pigs and reaching an average of 19.58 kg at 28 days after inoculation (Figure 6). Low variation in liveweight was detected between treatment groups at all timepoints with the largest variation seen at the end of the trial. The SFP and control groups had an increased (*p* < 0.05) average (with 95% CI) liveweight at this timepoint of 20.0 kg [19.8, 20.2] and 19.8 kg [19.6, 20.0], respectively, compared to 18.8 kg [18.5, 19.1] and 19.3 kg [18.1, 19.5] kg in the LAS and LFP groups, respectively.

Accordingly, average daily gain (ADG) increased over time with the mean ADG across all pigs being 81 g in week 0 (i.e., entry to day −7) and increasing to 220, 409, 518 and 620 g in subsequent weeks 1, 2, 3 and 4, respectively. Pigs fed diet SFP grew faster (565 vs. 473 g/day, *p* = 0.028) than pigs fed diet LAS between days 15 and 21 post-inoculation (Table 2).

Average daily feed intake (ADFI) increased over the duration of the study with the mean ADFI for all pigs being 123, 279, 509, 731 and 871 g for week 0 (i.e., entry to day −7), and subsequent weeks 1, 2, 3 and 4, respectively. Mean ADFI was similar (*p* > 0.05) between treatment groups in weeks 0, 1, 2 and 4, but in week 3, pigs fed diet SFP ate more feed than pigs fed diet LAS (835 vs. 662 g/day, *p* = 0.013). There were no differences (*p* > 0.05) in FCR detected between any of the dietary treatment groups during any time period (Table 2).

## 4. Discussion

Although many approaches in minimising the emergence of antimicrobial-resistant bacteria are being investigated and implemented in pork production, there is a lack of strategies to decolonise or reduce carriage of CIA-resistant bacteria once detected. The development of these strategies is necessary in minimising the risk of this resistance spreading between animals, farms and to humans. The current study aimed to establish an in vivo model to analyse the effects of postbiotic dietary supplements on reducing ESC-resistant *E. coli* in weaner pigs, while determining the effects of postbiotics on ESC-resistant *E. coli* shedding and performance.

The model that was established in this study successfully induced the shedding and potential colonisation of ESC-resistant *E. coli* over an extended timeframe. There was an initial increase in the shedding of ESC-resistant *E. coli* 24 h after challenge, with this concentration reducing by approximately two logs by 48 h post-challenge. Although the second peak in ESC-resistant *E. coli* demonstrated its replication and colonisation within the pigs, the small timeframe of high shedding concentrations may hinder identifying strategies to control resistant *E. coli*, with an ideal model invoking high shedding for a longer period. Despite breeding stock receiving a vaccination against *E. coli*, the ECOvac vaccine targets *E. coli* fimbrial antigens K88, K99 ad 987P, which were not present in the ESC-resistant *E. coli* challenge strain. Combined with the waning of maternal antibodies, it is unlikely this would have a confounding effect on the trial. Although this model provided insight into the dynamics of ESC-resistance, a repeat dosing schedule with ceftiofur or the use of an antimicrobial with a longer duration of activity in future studies may prolong the shedding of high levels of ESC-resistant *E. coli* via a reduction in competition from commensal gut flora. However, this would need to be carefully considered as a further confounding factor affecting gut stability.

Despite the short duration in high levels of ESC-resistant *E. coli* shedding, the persistence of low concentrations of ESC-resistant *E. coli* was also demonstrated in this study with 10% of pigs still harbouring ESC-resistant *E. coli* 28 days after challenge. Although at low concentrations (between 10^3^ and 10^4^ CFU/g), this persistence supports previous studies [9] and highlights the importance in reducing, and ultimately eliminating, ESC-resistant *E. coli*. Another point to consider in this model is the use of swabs for bacterial quantification instead of faecal samples. The collection of faecal samples is ideal and arguably provides more accurate data than swabs, but in the present study, faecal collection from 64 individual weaner pigs at multiple timepoints was considered unachievable. This was due to previous experience observing the sporadic time intervals between defecation in weaner pigs. In the current study, failure to detect *E. coli* only occurred in nine of the faecal swabs, supporting the use of swabs for sampling. An alternative solution would be to collect samples from a proportion of pigs as representatives for that treatment group. However, due to the relatively unstable gut microbiome in the period following weaning [20], this technique would most likely be unreliable in an experimental setting with low numbers of pigs. Despite the establishment of a successful experimental model, postbiotic supplementation with LFP and (or) SFP demonstrated no significant reduction (*p* < 0.05) in ESC-resistant *E. coli* and no general improvement in production in weaner pigs. This was most likely attributable to the high variation between pigs within a treatment, suggesting a larger sample size is required for future studies. Nevertheless, a positive trend (*p* = 0.065) of the SFP to reduce ESC-resistant *E. coli* carriage was demonstrated on day 14, compared to CON-fed pigs. In a previous study, broilers supplemented with SFP demonstrated a reduction in the virulence and resistance of the challenge *Salmonella* strain. This was attributed to the loss of the SGI1 integron [14]. Integrons are mobile genetic elements that can move intra- and inter-molecularly, meaning integrons can move within a DNA molecule or between DNA molecules and can insert into chromosomal or plasmid DNA [21]. In contrast, the current study challenged weaner pigs with a commensal strain confirmed to harbour the IncI1-*bla*_CTX-M-1_ plasmid. This plasmid is highly transferable, demonstrates long-term persistence in environments absent of direct selection pressures, and has been detected globally [6,9,10,22]. Coupled with its genetic similarity from isolates across continents and conservation of coding regions, the evidence suggests plasmids with an IncI1 backbone carrying the *bla*_CTXM-1_ gene are highly stable [9]. The stability of the plasmid, as well as the different bacterial strain that was used for challenge, may account for the contrasting results between these studies.

Another aspect to consider in future studies is the potential transfer of the plasmid to other commensal *E. coli* strains within the gastrointestinal tract. Genomic typing of ESC-resistant *E. coli* from multiple timepoints would determine if the plasmid had transferred to other strains, providing information on the transferability of the plasmid while verifying if all ESC-resistant *E. coli* quantified in the study was the original challenge strain. Although the postbiotics demonstrated no significant effect on ESC-resistant *E. coli* carriage in the current study, its previous success in boilers needs further exploration. Future exploration of the positive trend (*p* = 0.065) that the SFP postbiotics had on reducing ESC-resistant *E. coli* shedding is necessary using increased sample numbers and implementing mentioned strategies to increase the length of time in which high levels of ESC-resistant *E. coli* are shed. Furthermore, future studies using this model need to assess the effect of postbiotics against different ESC-resistant *E. coli* strains and the effect of postbiotics on resistance against different antimicrobials. Although this model can be used to evaluate control strategies through challenge with a single, commensal *E. coli* strain, natural carriage of *E. coli* is highly diverse with these postbiotics, and other alternate control strategies, requiring field-based trials [19,23]. The inclusion of pre- or postbiotics with these fermentation products may also increase the effects seen against the ESC-resistant *E. coli* with synbiotics, the combination of pre-, pro- or postbiotics, previously demonstrated to have a greater effect than the single additive [24]. Lastly, the concentration of postbiotics in the diet requires optimisation. Whereas there are many studies demonstrating the effects of varying concentrations of a single probiotic [25], in contrast, many of the studies investigating postbiotics investigate multiple postbiotics/combinations at a single concentration [26,27]. Determining the optimal concentration of the postbiotic may demonstrate an increased capability of the postbiotic against antimicrobial-resistant bacteria.

Supplementation with postbiotics demonstrated no statistically positive overall effects on growth performance, although some positive impacts of feeding SFP compared to LAS were noted in week 3 following inoculation. Studies investigating the effects of these postbiotics on the growth performance of swine are contrasting. Although LFP, SFP and the combination demonstrated increased growth performance in ETEC-challenged weaner pigs [17], other studies have demonstrated no significant effect on growth performance [28]. Meanwhile, Bass and Frank [29] reported an increased average daily gain in healthy weaner pigs. The beneficial effects of postbiotics have been demonstrated to be through modulation of the gut microbiome, and therefore the efficacy of postbiotics may be dependent on the microbiome of pigs prior to supplementation. In the current study, pigs were treated with ceftiofur before challenge with ESC-resistant *E. coli*, and due to the broad-range nature of antimicrobials, this treatment may have disrupted the gut microflora while also disrupting the effects of the postbiotics. Therefore the variation in the effects of postbiotics on growth performance may be attributed to the variation in the microbiome of pigs as impacted by environmental, host genetic factors and age [30]. Heightening our understanding of both postbiotics, the microbiome and their interactive relationship may allow increased consistency between studies and determination of the benefits of postbiotics in food-producing animals.

## 5. Conclusions

Overall, this study has demonstrated the applicability of an experimental model for analysing the effects of alternate control strategies against, but not restricted to, ESC-resistant *E. coli*. An in vivo model was used to assess the effects of postbiotics, in the form of LFP and SFP and their combination, against ESC-resistant *E. coli* levels in weaner pigs, with SFP demonstrating a positive statistical trend for a reduction in counts. The continued emergence and dissemination of ESC-resistant *E. coli* in livestock is a major One Health threat with the development of novel strategies that reduce resistance on farms urgently required to prevent its further dissemination.

## Figures and Tables

**Figure 1 animals-13-00959-f001:**
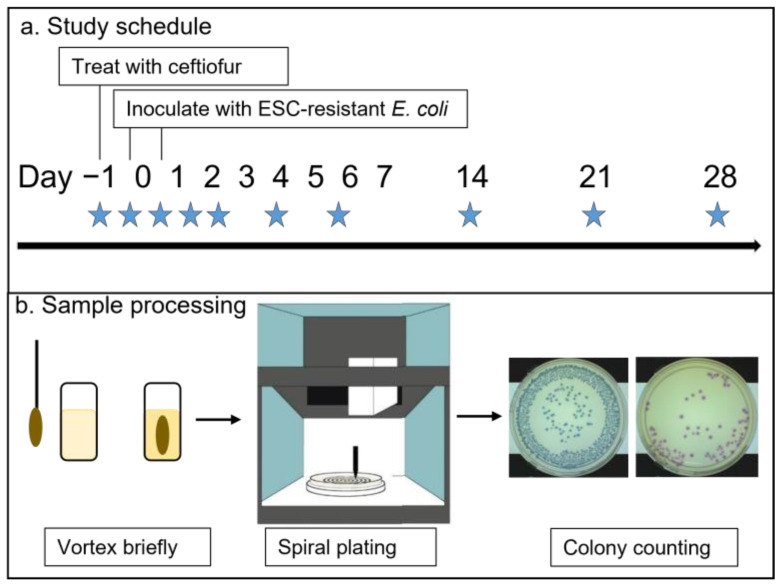
Overview of in vivo model for assessing effect of novel strategies on ESC-resistant *E. coli.* (**a**). Timeline of the study regarding treatments and sampling of pigs. Blue stars represent days rectal swabs were collected. The first inoculation with ESC-resistant *E. coli* was day 0 (i.e., 7 days after weaning). (**b**). Protocol of laboratory processing of samples for quantification of total *E. coli* and ESC-resistant *E. coli* from rectal swabs (calculated as CFU/g). Blue colonies represent *E. coli* on CHROMagar™ ECC agar while pink colonies represent ESC-resistant *E. coli* on CHROMagar™ ESBL agar.

**Figure 2 animals-13-00959-f002:**
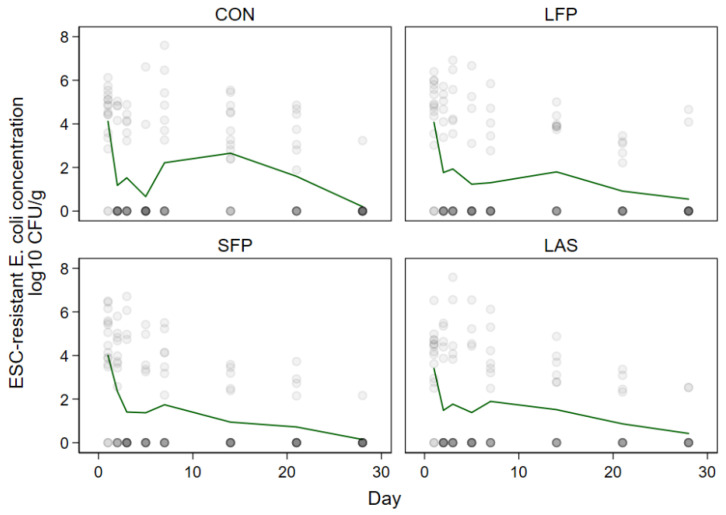
Mean concentration of ESC-resistant *E. coli* in pig rectal swabs (*n* = 16 per timepoint) belonging to different dietary treatment groups. Mean concentration is represented by the line with dots representing individual pigs (overlapping of individual pig data results in darker dots in graph). Treatment abbreviations: CON: control diet, LFP = CON + 2000 ppm *Lactobacillus acidophilus* fermentation product (LFP), SFP = CON + 2000 ppm *Saccharomyces cerevisiae* fermentation product (SFP), LAS = CON + 2000 ppm LFP + 2000 ppm SFP.

**Figure 3 animals-13-00959-f003:**
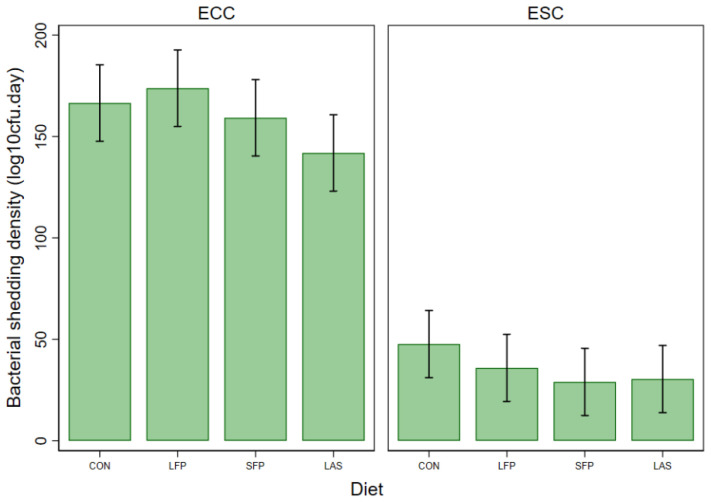
The total *E. coli* (ECC) and ESC-resistant *E. coli* (ESC) shedding density, a measure of bacterial shedding across the full duration of the trial, in pigs (*n* = 16 per timepoint) belonging to different dietary treatment groups. Error bars are standard error of the means. Treatment abbreviations: CON = control diet, LFP = CON + 2000 ppm *Lactobacillus acidophilus* fermentation product (LFP), SFP = CON + 2000 ppm *Saccharomyces cerevisiae* fermentation product (SFP), LAS = CON + 2000 ppm LFP + 2000 ppm SFP.

**Figure 4 animals-13-00959-f004:**
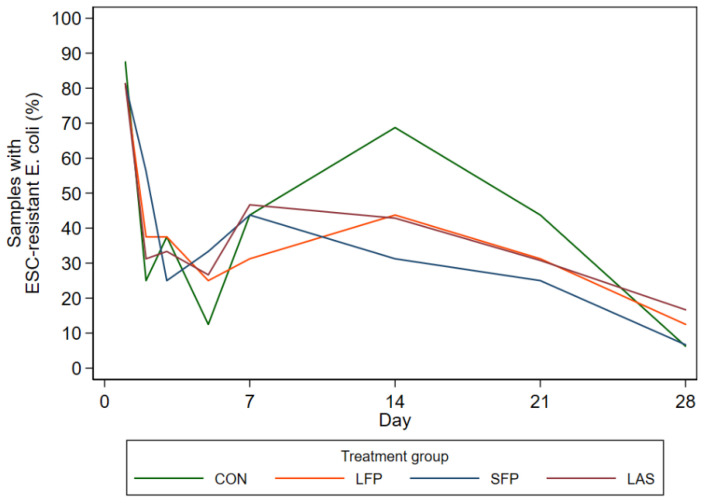
Percentage of pigs harbouring ESC-resistant *E. coli* over duration of the study and belonging to different dietary treatment groups. Treatment abbreviations: CON = control diet, LFP = CON + 2000 ppm *Lactobacillus acidophilus* fermentation product (LFP), SFP = CON + 2000 ppm *Saccharomyces cerevisiae* fermentation product (SFP), LAS = CON + 2000 ppm LFP + 2000 ppm SFP.

**Figure 5 animals-13-00959-f005:**
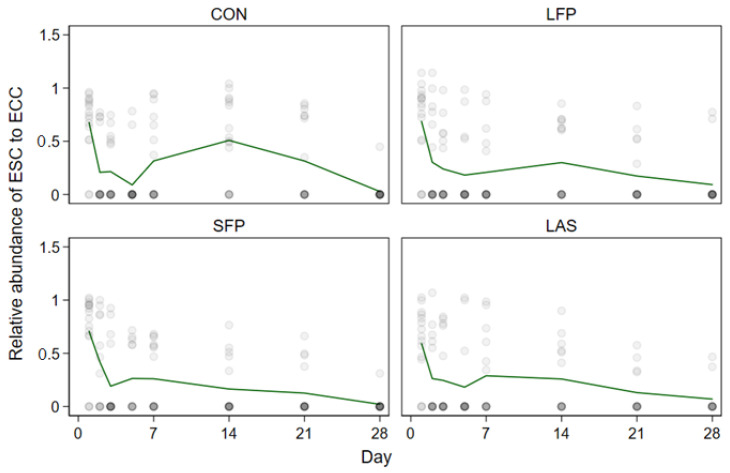
Abundance of ESC-resistant *E. coli* relative to total *E. coli* in rectal swabs from ESC-resistant *E. coli* challenged weaners belonging to different treatment groups. Line represents median with individual pigs represented by dots (overlapping of individual pig data results in darker dots in graph). Treatment abbreviations: CON = control diet, LFP = CON + 2000 ppm *Lactobacillus acidophilus* fermentation product (LFP), SFP = CON + 2000 ppm *Saccharomyces cerevisiae* fermentation product (SFP), LAS = CON + 2000 ppm LFP + 2000 ppm SFP.

**Figure 6 animals-13-00959-f006:**
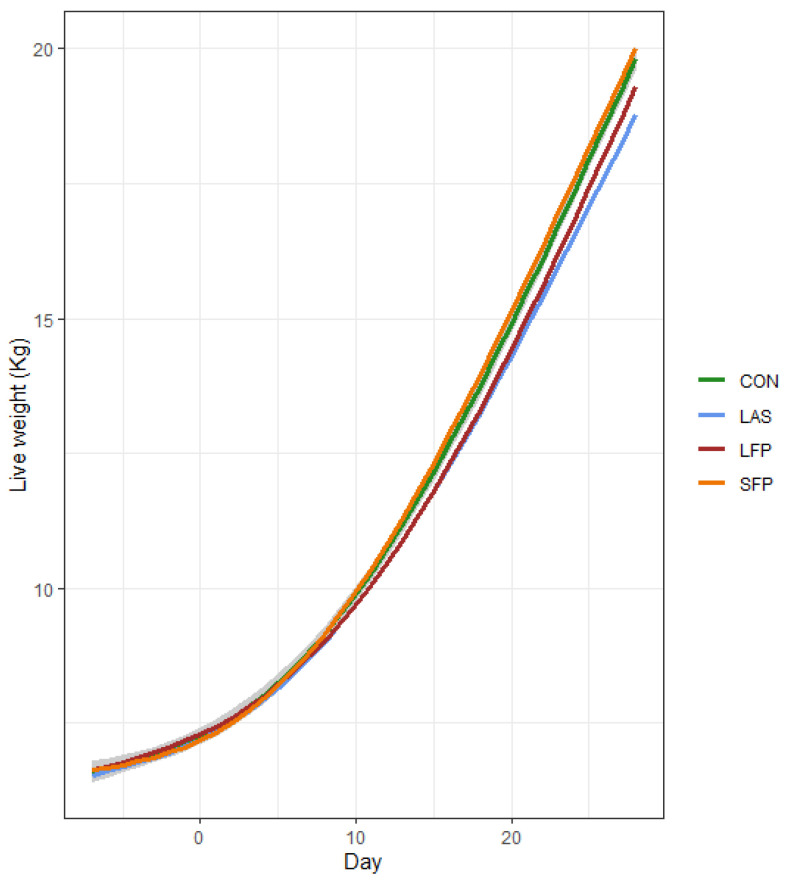
Mean liveweight of pigs (*n* = 416 per timepoint) in different dietary treatment groups in ESC-resistant *E. coli* challenged weaner pigs. Treatment abbreviations: CON = control diet, LFP = CON + 2000 ppm *Lactobacillus acidophilus* fermentation product (LFP), SFP = CON + 2000 ppm *Saccharomyces cerevisiae* fermentation product (SFP), LAS = CON + 2000 ppm LFP + 2000 ppm SFP.

**Table 1 animals-13-00959-t001:** A summary of the treatment used in the study.

Treatment	SynGenX, ppm	Diamond V Original XPC, ppm
CON	0	0
LFP	2000	0
SFP	0	2000
LAS	2000	2000

Treatment abbreviations: CON = control, LFP = *Lactobacillus acidophilus* fermentation product, SFP = *Saccharomyces cerevisiae* fermentation product, LAS = *Lactobacillus acidophilus* and *Saccharomyces cerevisiae* fermentation products.

**Table 2 animals-13-00959-t002:** Effects of postbiotics on average daily gain (ADG), average daily feed intake (ADFI) and the feed conversion ratio (FCR) in ESC-resistant *E. coli*-challenged weaner pigs.

Item	Treatment				
	CON	LFP	SFP	LAS	*p*-Value
ADG, g					
d −7 to 0	77 ± 14.3	85 ± 14.2	87 ± 16.9	75 ± 17.7	0.928
d 1–7	240 ± 16.6	205 ± 19.3	215 ± 24.9	222 ± 24.4	0.705
d 8–14	406 ± 27.3	393 ± 28.0	440 ± 19.0	396 ± 26.1	0.530
d 15–21	531 ± 19.4 ^ab^	493 ± 22.4 ^ab^	565 ± 22.5 ^a^	473 ± 25.6 ^b^	0.028
d 22–28	635 ± 22.9	639 ± 30.6	608 ± 36.0	593 ± 25.4	0.673
ADFI					
d −7 to 0	107 ± 10.9	129 ± 11.8	132 ± 18.4	121 ± 22.0	0.717
d 1–7	288 ± 20.4	280 ± 14.6	280 ± 28.6	266 ± 27.9	0.926
d 8–14	513 ± 27.7	507 ± 39.8	531 ± 29.3	485 ± 41.3	0.831
d 15–21	714 ± 29.9 ^ab^	713 ± 46.1 ^ab^	835 ± 27.3 ^a^	662 ± 12.0 ^b^	0.013
d 22–28	870 ± 27.6	898 ± 69.1	863 ± 33.0	852 ± 16.4	0.875
FCR					
d −7 to 0	1.7 ± 0.40	1.6 ± 0.19	1.6 ± 0.19	1.7 ± 0.08	0.996
d 1–7	1.2 ± 0.04	1.4 ± 0.05	1.3 ± 0.13	1.3 ± 0.10	0.596
d 8–14	1.3 ± 0.05	1.3 ± 0.04	1.2 ± 0.02	1.3 ± 0.04	0.263
d 15–21	1.3 ± 0.02	1.4 ± 0.03	1.5 ± 0.10	1.4 ± 0.08	0.495
d 22–28	1.4 ± 0.01	1.4 ± 0.05	1.3 ± 0.05	1.4 ± 0.04	0.315

^a,b^ Mean values within a row that have different superscripts are significant different (*p* < 0.05).

## Data Availability

The data presented in this study are available on request from the corresponding author.

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
