# Peer review of "Development of an In Vivo Extended-Spectrum Cephalosporin-Resistant Escherichia coli Model in Post-Weaned Pigs and Its Use in Assessment of Dietary Interventions"

_animals, 2023, doi:10.3390/ani13060959_

Round 1

Reviewer 1 Report

This paper aimed to assess if dietary supplementation with fermentation products of probiotic strains (postbiotics) reduced the development of antibiotic resistance in post-weaned pigs. Authors inoculated to pigs a cephalosporin-resistant strain (ESC-E. coli) and via fecal swabs, assessed the shedding of ESC-E. coli during a feeding with or without postbiotics. This manuscript describes an in vivo pig model and the trend of positive impact of one of the postbiotic tested, without statistically significant results.   

The subject of this manuscript fits the scope and the subject areas of the journal “Animals”. The manuscript is clear, well presented and well written. The experimental design is appropriate for the research objective. The data illustrations are properly and easy to understand. The results analysis and interpretation are appropriate and consistent. Discussion section is well documented and well supported. The authors are critical of their results and give suggestions for improving their model.

Specific minor comments:

Section 2.1: Pigs were housed 4 in each pen. So, is it right that there were 4 pens in each group (16 pigs in each feeding group)?

Table 1: Give the explanation of the treatment abbreviation (CON, LFP, SFP, LAS) below the table could improve the reading of the illustration.

Section 2.3: Rectal swabs were collected from each pig in each group, at each sampling day, isn’t it? Add from each pig after rectal swabs line 125.

Figures 2 and 5: several different color dots especially for value 0. Guess that it is due to the superposition of several negative samples? Please, clarify. Replace “faeces” with rectal swabs.

Figure 6: Please, indicate what is significant on the figure.

Section Discussion:  The authors could discuss about the concentration level of postbiotics that might have played a role in the study results. Are the postbiotics used in this study have been demonstrated a positive impact in vitro on the growth of ESC-resistant E. coli?  What about the alternative postbiotic + prebiotic?

Reviewer 2 Report

Major comments

Materials and Methods

L93-102: 

-        provide details about the selection criteria for selected pigs and commercial farms 

-        which was the mean BW of selected animals?

-        More details about the breed and vaccination program of commercial farm 

-        Were vaccinated the selected piglets?

-        Conclusions: you should expand the conclusions 

L104-122: 

-        Provide as a supplementary file the synthesis and analysis of the experimental diets

-         

Minor comments

§  L20, 26: antimicrobial-resistant

§  L24: using an increased sample size

§  L39: reducing the carriage of bacteria

§  L65: antimicrobial-resistant E. coli

§  L67: the fermentation of probiotic strains

§  L76: in ileal mucosa have been reported

§  L77: .. in pigs requires

§  L118: on days

§  L122: ..days

§  L138: .. with the distinction between

§  L144: .. ESC-resistant E. coli

§  L184: .. the shedding of ESC-resistant

§  L205: .. on days

§  L229: ..live weight..

§  L261: .. antimicrobial-resistant

§  L277: .. with a longer

§  L279: .. resistant E. coli via a reduction

§  L287: The collection of faecal samples

§  L335: .. on the growth performance

Reviewer 3 Report

Line 108. Please give the composition and nutritional level detailed information for the diet of each group and presented in one Table.

Line 176, 194, 242. There is a wrong description for n = 64, maybe it was n = 4. Because the repetition number of each group is 4.

Figure 2 and Figure 5. The combined figure with the results of all the groups also needed to be given, which is similar to figure 4. In addition, the results presented with statistical analysis are also needed.

After weaner piglets were inoculated with ESC-resistant E. coli. Whether there will be diarrhea?
